# Characteristics and Research Techniques Associated with the Journal Impact Factor and Other Key Metrics in Pharmacology Journals

**Mingkwan Na Takuathung** [1] 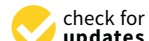, **Wannachai Sakuludomkan** [1], **Supanimit Teekachunhatean** [1] **and Nut Koonrungsesomboon** [1,2,*]

[1] Department of Pharmacology, Faculty of Medicine, Chiang Mai University, Chiang Mai 50200, Thailand; mingkwan.n@cmu.ac.th (M.N.T.); wannachai.yin@gmail.com (W.S.); supanimit.t@cmu.ac.th (S.T.)

[2] Musculoskeletal Science and Translational Research (MSTR) Center, Chiang Mai University, Chiang Mai 50200, Thailand

\* Correspondence: nut.koonrung@cmu.ac.th; Tel.: +66-5393-5353

**Abstract:** In the present age, there is intense pressure on researchers to publish their research in 'high-impact factor' journals. It would be interesting to understand the trend of research publications in the field of pharmacology by exploring the characteristics of research articles, including research techniques, in relation to the journal's key bibliometrics, particularly journal impact factor (JIF), the seemingly most mentioned metric. This study aimed to determine the characteristics and research techniques in relation to research articles in pharmacology journals with higher or lower JIF values. A cross-sectional study was conducted on primary research journals under the 'Pharmacology and Pharmacy' category. Analysis of 768 original research articles across 32 journals (with an average JIF of 2.565 ± 0.887) demonstrated that research studies involving molecular techniques, in vivo experiments on animals, and bioinformatics and computational modeling were significantly associated with a higher JIF value of the journal in which such contributions were published. Our analysis suggests that research studies involving such techniques/approaches are more likely to be published in higher-ranked pharmacology journals.

**Keywords:** altmetric; bibliometrics; impact factor; pharmacology; research methodology

## 1. Introduction

Journal Impact Factor (JIF) is arguably the most popular journal metric, and it has been widely used as a proxy indicator of journal prestige for over half a century [1,2]. Despite the limitations addressing it from different perspectives, the JIF is quantifiable, making it a metric that could differentiate the overall performance of a journal at a glance [3,4]. The JIF analyzes the average number of citations received per paper published in a journal during the preceding two years [5]. For instance, the 2019 JIF of 2.51 indicates that each published article in 2017 and 2018 generated, on average, 2.51 citations by the end of 2019. Thus, the JIF is ordinarily viewed as a predictor of the future citation rate of published articles, although that it is not necessarily the case [6,7].

Apart from JIF, other alternative metrics have been developed to measure journal performance and research quality from different outlooks [8,9]. For instance, Eigenfactor Score (ES) is proposed to reflect the influence of journals and their published articles; the ES calculation gives more weight to the citations from higher-ranked journals than those from lower ones [10]. Article influence score is a derivative of ES; it accounts for the average influence of an article in a journal, which is directly comparable to the JIF [11]. An article influence score greater than 1 indicates that the average article in the journal has above-average influence within the database. Although those metrics are not as widely used as JIF, they give complementary information on the overall quality and impact of the journal regarding its published articles [12–14].

In the current era, there is intense pressure on researchers to publish their research in 'high-impact factor' or prestigious journals [15,16]. Under the contemporary academic evaluation systems [17], it is undeniable that many researchers customarily use the JIF and/or other metrics as a determining factor for submission of their manuscript to a journal [18,19]. Thus, it is of interest to explore the characteristics of original research articles, including research techniques being used, in relation to the journal's key bibliometrics, such as JIF and article influence score, to understand the trend of research publications in particular fields of interest [20]. Therefore, the objective of the present study was to determine the association between the characteristics of published original research articles and the journal's key metrics within the field of pharmacology.

## 2. Materials and Methods

### 2.1. Study and Design

In this cross-sectional study, the scientific journals under the "Pharmacology and Pharmacy" category in the 2019 ISI® Journal Citations Report (JCR) were retrieved on 1 July 2020. The characteristics of the journal to be included in this study were as follows: (1) the journal does have the term "pharmaco" in its journal name, (2) the journal published citable items in 2019 of more than 90%, among which more than 90% are original research articles, and (3) the journal published original research articles of more than 50 articles in 2019. With the above-mentioned criteria, a total of 33 journals were eligible for inclusion. We could not get access to full-text articles of one journal, so a total of 32 journals were included in this study.

All articles of the 32 journals published in 2019 were retrieved. An initial screening of the articles was performed to include only original research articles. Narrative review articles, case reports, case series, editorials, letters to the editor, commentaries, correspondences, and expert opinions were excluded. We randomly selected 24 original research articles from each journal (approximately 2 articles published in each month) using a computer-generated random-number sequence. The sample size of 24 articles per journal was deliberate to obtain a 10% absolute precision and 90% confidence based on the assumption of that 9% of original research articles would have each research technique parameter.

Data to be extracted in a standardized form included (1) authors/affiliations (number of authors, number of affiliations, country income level of the corresponding author, and h-index of the corresponding author), (2) research fields, and (3) research techniques or approaches (e.g., molecular techniques, omics techniques, bioinformatics and computational modeling, docking and simulation, meta-analysis, in vivo experiments on animals, ex vivo experiments on isolated tissues from living organisms, interventions on human subjects, observations involving human subjects, use of a questionnaire, and use of a registry database). The 2019 JIF and other key metrics of each journal (including 5-year JIF, JIF percentile, normalized ES, article influence score, and immediacy index) were obtained from the 2019 JCR on 1 July 2020.

### 2.2. Statistical Analysis

Descriptive analyses were presented as the frequency with percentage or mean with standard deviation, as appropriate. Spearman correlation was applied to explore the strength of relationships between two different metrics. Associations between journal metrics and characteristics of original research articles and research techniques were examined using multiple linear regression analysis. The JIF and article influence score were treated as primary and secondary dependent variables (endpoints), respectively, while the characteristics of original research articles and research techniques were treated as independent variables. Each research technique was handled as one dichotomous variable (0 = no; 1 = yes) when it was included in the multiple linear regression analysis model. Statistical analyses were performed using SPSS (IBM Corp. Released 2013. IBM SPSS Statistics for

Windows, Version 22.0. Armonk, NY, USA). All statistical tests were two-sided, and a *p*-value of less than 0.05 was regarded to indicate statistical significance.

### 3. Results

A total of 32 journals were eligible for inclusion, 12 of which (37.5%) were published by Elsevier and 4 (12.5%) by Wiley. The JIF value of the 32 journals ranged from 0.692 to 5.064, with the mean JIF value of 2.565 ± 0.887. The average article influence score was 0.587 ± 0.229, while the mean values of other key metrics are shown in Table 1. Spearman's rank correlation coefficient among key metrics is presented in Figure 1; all the metrics had moderate-to-high correlation among one another.

**Table 1.** Journal impact factor and other key metrics of the included journals.

| Indicators | Mean | SD | Median | Range |
|---|---|---|---|---|
| JIF | 2.565 | 0.887 | 2.667 | 0.692–5.064 |
| Five-year JIF | 2.578 | 0.853 | 2.615 | 0.893–4.588 |
| JIF percentile | 46.559 | 20.663 | 50.635 | 4.259–83.555 |
| Normalized ES | 0.85404 | 0.80972 | 0.54097 | 0.02773–3.18115 |
| Article influence score | 0.587 | 0.229 | 0.583 | 0.127–1.145 |
| Immediacy index | 0.633 | 0.306 | 0.573 | 0.154–1.525 |

JIF, Journal Impact Factor; ES, Eigenfactor Score.

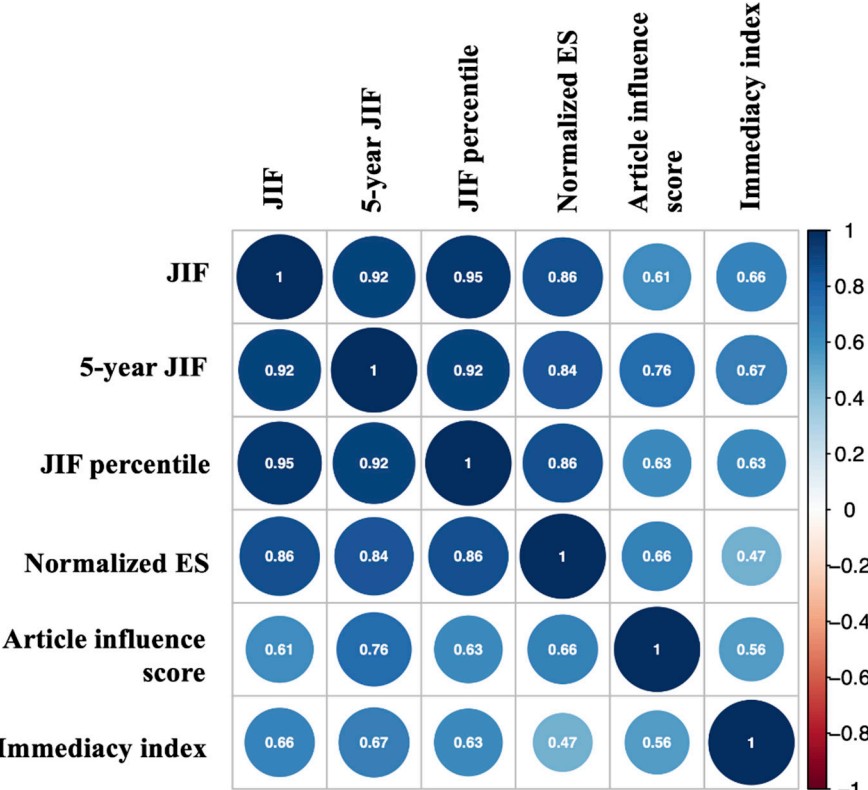

**Figure 1.** Spearman's rank correlation coefficient for the included journals. All the correlations are significant at the 0.01 level (2-tailed). JIF, Journal Impact Factor; ES, Eigenfactor Score.

A total of 768 articles across 32 journals (24 articles per journal) were retrieved and included in our analysis. Each article had an average of 7.0 ± 3.3 authors and 3.5 ± 2.2 affiliations. The majority of included articles had the main corresponding author affiliated

with an institution or university in China ($n$ = 204, 26.6%) or United States ($n$ = 140, 18.2%), followed by Japan ($n$ = 53, 6.9%), Turkey ($n$ = 35, 4.6%), Italy ($n$ = 32, 4.1%), and Korea ($n$ = 31, 4.0%). More than nine-tenths of the articles came from countries with upper-middle to high-income levels according to the 2020 World Bank classification. The h-index of the main corresponding author was 16.5 ± 13.2. The majority of research articles were in the field of neuropharmacology or psychopharmacology ($n$ = 245, 31.9%), inflammation, immunomodulation or hematopoiesis ($n$ = 156, 20.3%), and cardiovascular pharmacology, pulmonary pharmacology, or renal pharmacology ($n$ = 139, 18.1%). Around three-fourths of the articles (73.3%) involved in vitro experiments, while 43.8%, 38.0%, and 15.6% involved animal models, human subjects, and in silico approaches, respectively (Table 2).

**Table 2.** Characteristics of the included articles.

| Variables | Mean ± SD (Range) or Frequency (%) |
|---|---|
| Number of authors | 7.0 ± 3.3 (1–20) |
| Number of institutions | 3.5 ± 2.2 (1–19) |
| Main corresponding author | |
| *h*-index | 16.5 ± 13.2 (0–102) |
| Country income levels | |
| High | 418 (54.4%) |
| Upper middle | 310 (40.4%) |
| Lower middle | 38 (4.9%) |
| Low | 2 (0.3%) |
| Research fields | |
| Neuropharmacology or psychopharmacology | 245 (31.9%) |
| Inflammation, immunomodulation, or hematopoiesis | 156 (20.3%) |
| Cardiovascular pharmacology, pulmonary pharmacology, or renal pharmacology | 139 (18.1%) |
| Pharmacotherapy of neoplastic disease | 104 (13.5%) |
| Gastrointestinal pharmacology | 75 (9.8%) |
| Hormones and hormone antagonists | 72 (9.4%) |
| Chemotherapy of infectious diseases | 46 (6.0%) |
| Miscellaneous | 142 (18.5%) |
| Approaches | |
| Human studies | 292 (38.0%) |
| Animal studies | 336 (43.8%) |
| In vitro studies | 563 (73.3%) |
| In silico studies | 120 (15.6%) |

In the multiple linear regression analysis, five independent characteristics/research techniques, i.e., molecular techniques, in vivo experiments on animals, bioinformatics and computational modeling, number of authors, and h-index of the corresponding author, were found to be significantly associated with a higher JIF (Table 3). The model was also applied to identify characteristics of the articles in relation to article influence score. It was observed that a higher article influence score was significantly associated with 10 independent characteristics/research techniques, four of which (i.e., molecular techniques, in vivo experiments on animals, number of authors, and h-index of the corresponding author) were similar to the analysis of the primary endpoint (see Table S1).

**Table 3.** Associations between study characteristics/research techniques and journal impact factor.

| Variables | B | 95% Confidence Interval | SE | Beta | *p* Value |
|---|---|---|---|---|---|
| Molecular techniques | 0.377 | 0.252 to 0.503 | 0.064 | 0.195 | <0.001 |
| In vivo experiments on animals | 0.366 | 0.252 to 0.480 | 0.058 | 0.207 | <0.001 |
| Bioinformatics and computational modeling | 0.290 | 0.046 to 0.533 | 0.124 | 0.078 | 0.020 |
| Number of authors | 0.059 | 0.041 to 0.077 | 0.009 | 0.220 | <0.001 |
| *h*-index of the corresponding author | 0.010 | 0.006 to 0.015 | 0.002 | 0.126 | <0.001 |

B, unstandardized beta (representing the slope of the line between the independent variable and the dependent variable); SE, standard error; Beta, standardized beta (representing the strength and direction of the relationship between the independent variable and the dependent variable).

## 4. Discussion

To our knowledge, the present study is the first systematic analysis determining the characteristics and research techniques being employed in relation to published original research articles in pharmacology journals with higher or lower JIF. Our analysis suggests that research studies involving molecular techniques, in vivo experiments on animals, and/or bioinformatics and computational modeling are more likely to be published in pharmacology journals with higher JIF. It is reasonable to assume that molecular techniques often bring about some novel findings on a mechanism of drug action, leading to the results being published in higher-ranked journals [21,22]. In vivo experiments on animals are suited for observing or testing the overall effects of a drug on living organisms [23], while bioinformatics and computational modeling techniques hold great promise for the advancement of pharmacology research in drug design, pharmacokinetics, and drug action [24,25]. In consequence, research studies involving such techniques/approaches are more likely to be accepted for publication in higher-ranked journals. Our analysis also observed the relationship between the number of authors and *h*-index of the corresponding author and the JIF value. This finding may imply that higher-ranked journals tended to publish research studies involving multidisciplinary collaboration or those conducted by renowned researchers [26].

It is to be noted that the present study does not intend to associate the JIF itself with the quality of individual research articles, but rather to determine any associations of research characteristics and/or techniques with the JIF and other metrics. The JIF has evolved from its originally intended use to help us understand how influential a journal is; it is not a magic number that can straightforwardly epitomize the quality of scientific content in each research article [27,28]. In other words, a high JIF value does not necessarily reflect the high quality of individual research articles published in the journal and may not even be related to accrued citations of the articles [29,30]. It is widely acknowledged that a high JIF value of some journals may be derived primarily from a small percentage of high-impact articles accounting for the majority of citations of the journal [31–35]. Indeed, the quality and impact of any research studies should be judged on scientific merit and/or social value rather than on any bibliometrics alone [19,36–38].

The results of the present study should be treated with caution. First, it must be acknowledged that our analysis was limited to original research articles published in the pharmacology journals under the "Pharmacology and Pharmacy" category. It is rather common that research articles on clinical pharmacology or pivotal drug trials are published in general medical journals or multidisciplinary journals with a high JIF value [39]. Besides, pharmacogenomics studies involving large genomic datasets are frequently published in high-impact journals under the "Genetics & Heredity" category [40]. Since any of these given examples were not included and did not figure in our analysis, it should be kept in mind that the analysis might misestimate or fail to identify some associations between certain techniques/approaches and the JIF value.

Second, our analysis was limited to primary research journals (defined as "any journals that publish citable items of more than 90% and among which more than 90% are

original research articles"). Consequently, certain highly cited pharmacology journals that published several review articles and other types of articles were excluded from the analysis. It is possible that original research articles published in such journals might have had different characteristics from what we observed in the primary research journals. However, the present study was designed to include only the primary research journals in order to minimize the effects of inherent biases in the JIF calculation on the study endpoints [41]. It is widely acknowledged that JIF calculations are highly vulnerable to manipulations by increasing the number of non-citable items (e.g., editorials and commentaries) which may be cited but are not counted in the denominator of the JIF calculation [42,43]. Furthermore, review articles are likely to be cited more frequently than original research articles, so they can inflate the JIF of a journal [44,45].

## 5. Conclusions

In summary, research studies involving molecular techniques, in vivo experiments on animals, and/or bioinformatics and computational modeling were significantly associated with a higher JIF value of the journal in which such original contributions were published. Our analysis indicates that higher-ranked pharmacology journals place a priority on original research studies involving such techniques/approaches for the advancement of knowledge in the field of pharmacology. In other words, research studies involving such techniques/approaches are more likely to be published in pharmacology journals with a higher JIF value.

**Supplementary Materials:** The following are available online at https://www.mdpi.com/article/10.3390/computation9110116/s1, Table S1. Associations between study characteristics/research techniques and article influence score.

**Author Contributions:** N.K. and S.T. conceived the study; N.K., M.N.T., and W.S. designed the study; M.N.T. and W.S. collected the studies and extracted the data; N.K. and M.N.T. analyzed the data; N.K. wrote the paper, with contributions from all authors. All authors have read and agreed to the published version of the manuscript.

**Funding:** The authors received no specific funding for this work.

**Data Availability Statement:** The data that support the findings of this study are available on request from the corresponding author.

**Conflicts of Interest:** The authors declare no conflict of interest.

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
