# Peer review of "Characteristics and Research Techniques Associated with the Journal Impact Factor and Other Key Metrics in Pharmacology Journals"

_computation, doi:10.3390/computation9110116_

Round 1
Reviewer 1 Report
This is an excellent analysis of how specific research subjects tend to be published in journals with higher IF in the field of ‘Pharmacology and Pharmacy’.
Author Response
We thank the reviewer for the careful and insightful review of our manuscript. Our responses to the comments are described below in a point-by-point manner. Thank you for your consideration.
Reviewer’s Comment:
This is an excellent analysis of how specific research subjects tend to be published in journals with higher IF in the field of ‘Pharmacology and Pharmacy’.
Authors’ Reply:
Thank you very much for your positive comments.
Reviewer 2 Report
The paper "Characteristics and Research Techniques Associated with the Journal Impact Factor and Other Key Metrics in Pharmacology Journals" is a useful attempt to determine the characteristics and research techniques in relation to research articles in pharmacology journals with higher or lower JIF. The chosen methods and stages of the study are logical and clear, the authors have properly conducted a statistical analysis. The results are presented in an informative figure and tables, and the conclusions and discussions logically follow from the results. I should also mention a good list of references in the paper.
I have a few brief comments and suggestions:
The Journal Citations Report data for this study were obtained still last year (July 1, 2020). However, the presented systematic analysis is valuable primarily for its methodology, and it is important that this analysis can be repeated for other fields and years.
This analysis shows the priorities of higher-ranked pharmacology journals in the past. But can we say that such trends will continue in the future? The authors rightly note in the abstract: “Our analysis suggests that research studies involving molecular techniques, in vivo experiments on animals, and/or bioinformatics and computational modeling are more likely to be published in pharmacology journals with higher JIF”. Perhaps this should also be emphasized in the conclusions.
In the recent study, researchers have studied the trends of high-impact studies in pharmacology and pharmacy research https://doi.org/10.3389/fphar.2021.726668 The authors used completely different methodologies, but since both of these works relate to high-impact studies in pharmacology and pharmacy research, it may be worth mentioning this new study in this paper.
Please note that the comments and suggestions I have made here do not affect the overall positive assessment of this paper.
Author Response
We thank the reviewer for the careful and insightful review of our manuscript. We sincerely appreciate all the valuable comments and suggestions, which help us improve the quality of our manuscript. Our responses to the comments are described below in a point-by-point manner. Appropriate changes, suggested by the reviewer, have been added to the manuscript. Thank you for your consideration.
Reviewer’s Comment:
This is an excellent analysis of how specific research subjects tend to be published in journals with higher IF in the field of ‘Pharmacology and Pharmacy’.
Authors’ Reply:
Thank you very much for your positive comments.
Reviewer’s Comment:
The paper “Characteristics and Research Techniques Associated with the Journal Impact Factor and Other Key Metrics in Pharmacology Journals” is a useful attempt to determine the characteristics and research techniques in relation to research articles in pharmacology journals with higher or lower JIF. The chosen methods and stages of the study are logical and clear, the authors have properly conducted a statistical analysis. The results are presented in an informative figure and tables, and the conclusions and discussions logically follow from the results. I should also mention a good list of references in the paper.
Authors’ Reply:
Thank you very much for your positive comments.
Reviewer’s Comment:
I have a few brief comments and suggestions:
The Journal Citations Report data for this study were obtained still last year (July 1, 2020). However, the presented systematic analysis is valuable primarily for its methodology, and it is important that this analysis can be repeated for other fields and years.
Authors’ Reply:
Thank you very much for your suggestion. A repeated analysis is planned to be done in the future to see whether the trend is changed or not.
Reviewer’s Comment:
This analysis shows the priorities of higher-ranked pharmacology journals in the past. But can we say that such trends will continue in the future? The authors rightly note in the abstract: “Our analysis suggests that research studies involving molecular techniques, in vivo experiments on animals, and/or bioinformatics and computational modeling are more likely to be published in pharmacology journals with higher JIF”. Perhaps this should also be emphasized in the conclusions.
Authors’ Reply:
Thank you very much for your suggestion. We have added what the reviewer suggests in the conclusions of the revised manuscript.
Reviewer’s Comment:
In the recent study, researchers have studied the trends of high-impact studies in pharmacology and pharmacy research https://doi.org/10.3389/fphar.2021.726668 The authors used completely different methodologies, but since both of these works relate to high-impact studies in pharmacology and pharmacy research, it may be worth mentioning this new study in this paper.
Authors’ Reply:
Thank you for your suggestion. This recent article is interesting and we have cited it in our revised manuscript.
Reviewer’s Comment:
Please note that the comments and suggestions I have made here do not affect the overall positive assessment of this paper.
Authors’ Reply:
Thank you for your valuable comments and suggestions.